# Higher Intake of Fat, Vitamin E-(β+γ), Magnesium, Sodium, and Copper Increases the Susceptibility to Prostatitis-like Symptoms: Evidence from a Chinese Adult Cohort

**DOI:** 10.3390/nu14183675

**Published:** 2022-09-06

**Authors:** Meng Zhang, Chen Jin, Yang Ding, Yuqing Tao, Yulin Zhang, Ziyue Fu, Tao Zhou, Li Zhang, Zhengyao Song, Zongyao Hao, Jialin Meng, Chaozhao Liang

**Affiliations:** 1Department of Urology, The First Affiliated Hospital of Anhui Medical University, Hefei 230022, China; 2Institute of Urology, Anhui Medical University, Hefei 230022, China; 3Anhui Province Key Laboratory of Genitourinary Diseases, Anhui Medical University, Hefei 230022, China; 4The Second Clinical Medical College, Anhui Medical University, Hefei 230022, China; 5The First Affiliated Hospital of Anhui Medical University, Hefei 230022, China; 6Anhui Clinical Research Center of Urology Disease, Hefei 230022, China

**Keywords:** prostatitis-like symptoms, micronutrients, Chinese adult, susceptibility

## Abstract

Background: Prostatitis-like symptoms (PLS) lead to severe discomfort in males in their daily lives. Diet has been established as affecting PLS in our prior study, but the effect of nutrients, particularly for micronutrients remains largely unclear. Methods: This study enrolled 1284 participants from August 2020 to March 2021. The National Institute of Health–Chronic Prostatitis Symptom Index was used to assess PLS. The diet composition was evaluated by the Chinese Food Composition Tables. Results: Participants were separated into PLS (*n* = 216), control (*n* = 432), and noninflammatory-abnormal symptoms (NIANS) (*n* = 608) groups. We observed higher levels of carotene, vitamin C, vitamin E-(β+γ) and subclass, zinc, magnesium, selenium, potassium, sodium, iron and manganese in the PLS group than in the control group. After adjustment for the potential confounders, the elevated risk from IQR2 to IQR4 of fat (*P _for trend_* = 0.011), vitamin E-(β+γ) (*P _for trend_* = 0.003), magnesium (*P _for trend_* = 0.004), sodium (*P _for trend_* = 0.001) and copper (*P _for trend_* < 0.001) was identified. Conclusions: This is the first study to evaluate the nutrient distribution in PLS patients and reveal that the higher intake of fat, vitamin E-(β+γ), magnesium, sodium, and copper is associated with a risk of PLS.

## 1. Introduction

The prevalence of prostatitis-like symptoms (PLS), which is regarded as a series of symptoms of chronic prostatitis/chronic pelvic pain syndrome (CP/CPPS), including the urgency to urinate, painful or difficult urination, painful ejaculation, pain in the abdominal or pelvic region and abnormal blooding in urine or semen, is 2–14% in males [1,2,3,4]. The influence of diet on the pathogenesis of PLS has been widely investigated in recent years. Over the years, alcohol consumption and spicy food, regardless of low-vegetable or meat-rich diets, have been reported to increase the risk of PLS [5,6,7].

Current evidence suggests that infection, defective urothelial integrity function, autoimmunity, or even psychosocial status significantly affect CP/CPPS [8]. Immune system imbalance has been extensively investigated in studies on CP/CPPS; the activation of Th1 and Th17 lymphocytes can reportedly lead to the secretion of pro-inflammatory cytokines, including IFN-γ and IL-17A, resulting in the interaction of pathogenic effects, maintaining and amplifying the immune-mediated inflammatory alteration in the prostate gland [9]. Moreover, CD25^+^ Treg cells could inhibit Th1/cytotoxic T-cell 1 (Tc1) autoimmune interaction and decrease the level of CXCR3 in T-effector cells, negatively affecting the homing of T-effectors to the prostate and leading to the inhibition of chronic prostatitis pathological changes [10]. Notably, every stage of the immune response is determined by the presence of certain micronutrients, and micronutrients are essential to the immune system. It has been established that vitamin E maintains or enhances NK cell cytotoxic activity [11,12,13] and inhibits PGE2 production in macrophages (thus indirectly protecting T-cell function) [11,14]. Magnesium participates in antigen binding to macrophages [15], regulating leukocyte activation [15], and is involved in the regulation of apoptosis [16]. Copper has been reported to accumulate at sites of inflammation [17,18] and is a constituent of copper/zinc-superoxide dismutase, a key enzyme in defense against reactive oxygen species (ROS) [19]. Although micronutrients are fundamental to immune function, excessive intake can lead to immune hyperfunction. Reducing micronutrient supplementation may be beneficial in such cases in order to reduce the susceptibility to PLS.

The purpose of this study was to comprehensively examine the role of micronutrients in susceptibility to PLS via collecting information on daily dietary intake from a population-based case-control study.

## 2. Materials and Methods

### 2.1. Participants Summary

We enrolled participants from August 2020 to March 2021, mainly in Hefei, Anhui Province, China. Participants (including the controls and cases) were recruited from local hospitals, nearby universities, and communities. Most participants in the case group were from the urology clinic, and a small percentage were from nearby universities or communities, and whose chief complaint was pain or discomfort in the perineum and/or during ejaculation. Correspondingly, we randomly recruited subjects without PLS from the same locations into the control group.

### 2.2. Inclusion and Exclusion Criteria

The National Institute of Health–Chronic Prostatitis Symptom Index (NIH-CPSI) is broadly used to estimate PLS. Modifications were made according to Nickel et al.’s [20] and our clinical experience; participants that complained of pelvic/perineal zone and/or ejaculatory pain or discomfort, with pain domain scores of National Institutes of Health chronic prostate symptom index (NIH-CPSI) ≥ 4, were assigned to the PLS group. The participants with an NIH-CPSI score of zero were assigned to the control group, and participants not included in the PLS group but with any scores for NIH-CPSI were assigned to the noninflammatory-abnormal symptoms (NIANS) group. In addition, participants under 18 years old or with acute genitourinary infection and pelvic surgery within three months were excluded.

### 2.3. Variables Definition and Data Collection

The clinical outcome was regarded as the dependent variable in the current study, which was quantified by the NIH-CPSI questionnaire, and final subgroups of control, PLS and NIANS were separated via the above-mentioned criteria.

We recorded the subjects’ demographic data as covariates, which have been reported as susceptible factors of PLS in our previous studies [7], including BMI, age, economic pressure, and educational level. BMI was calculated by dividing the weight of the subjects (kilogram) by the square of height (meters), and economic pressure was divided into none, mild, moderate and severe. The Chinese education system consists of six years of primary school, three years of junior high school, three years of senior high school, and is followed by masters, bachelors and doctoral degrees. In the current study, we separated the participants into three subgroups: junior high school or below, senior high school or bachelor’s degrees, and master’s degree or above.

For the average daily intake of micronutrients, a customized questionnaire was used to record the daily diet information (Appendix A), including dairy products, seafood, beans, meat, eggs, cereals, vegetables, nuts, fruit, etc. We evaluated the average daily dietary intake (gram, g) by asking subjects to recall the intake frequency and content of each listed food. Foods with high intake frequency were recorded in the form of how many times per week or even per day they were consumed, while low-intake food frequency was recorded per month. For foods with extremely low intake frequency (an average intake of less than once per month), the intake was recorded as zero. As for some foods pointed out by subjects to be seasonal, their intake frequency was recorded in the same way as above but further divided by four. In addition, when the subjects did not clearly recall the specific frequency of the intake of a certain food, we recorded the median value. For example, if the subject recalled consuming a food one to two times per week, it was recorded as 1.5 times per week. A validated retrospective dietary survey chart was used to reduce the recall bias and accurately quantify each dietary intake [21]. Furthermore, to calculate the average daily intake of micronutrients, the content of per 100 g in commonly consumed food was compiled, based on the Chinese Food Composition Tables [22] (Appendix A).

The data collection was conducted with the single-blind method; the informed consent form, NIH-CPSI, and homemade questionnaire were recorded separately, which helps to obtain authentic and valid sensitive information and reduces exposure suspicion bias. All data were obtained through offline one-to-one surveys, and investigators received adequate training.

### 2.4. Statistical Analyses

The non-parametric quantitative data were expressed as the median value with interquartile range (IQR), and comparisons between groups were conducted by Kruskal-Wallis (K-W) test. The qualitative data were displayed as the percentage, and comparisons between groups were conducted by the Chi-square test. Logistic regression analysis was conducted to compute the odds ratio (OR) and 95% confidence interval (CI) to estimate the association between the nutrients with PLS. Data for each nutrient were divided into four groups; IQR1 was chosen as the reference group in logistic regression analysis to reveal the impact of the three other groups. The covariates with significant differences between groups were included and displayed as different models. Among them, age and BMI were included in the model as continuous variables, while educational level and economic pressure were included in the model as categorical variables. We estimated the variations in the regression coefficients of variables in different models before and after the adjustment for potential confounders, such as total daily diet-energy intake or age. To reveal the gradual influence of the increasing factor weight, we calculated the *p* for trend value, which reportedly displays the linear trend of the ORs. Part of statistical analyses were performed on Extreme Smart Analysis website (https://www.xsmartanalysis.com/, accessed on 31 August 2022) and all statistical analyses were performed using SPSS Statistics 25.0 (IBM Corp., Armonk, NY, USA) and R4.1.2. A two-tailed *p*-value < 0.05 was statistically significant.

## 3. Results

### 3.1. Basic Information of Enrolled Participants

We initially collected a total of 1284 questionnaires. After excluding subjects missing important data (*n* = 28), 1256 participants were finally included for subsequent analysis. Among them, 216 participants complained of pain or discomfort in the pelvic or perineal area or during ejaculation in the past week and were assigned to the PLS group, while 432 participants with no symptoms of prostatitis were attributed to the control group, and the remaining participants with little or no pain symptoms associated with prostatitis (<4 points), but experiencing several symptoms from the NIH-CPSI, such as frequent micturition or urgent urination were assigned to the NIANS group (*n* = 608) (Figure 1). For the demographic data, we observed different distributions for age and economic pressure between the control and PLS groups but not between the control and NIANS groups. The PLS group presented a higher age and lower education level than the control and NIANS groups. The difference in BMI and economic pressure among the three groups was not significant (Table 1). The above-mentioned factors were potential confounders and were adjusted for in the subsequent analysis.

### 3.2. Identification of Independent Risk Factors of PLS via Different Logistic Regression Models

To identify independent risk factors for PLS, we performed a logistic regression analysis between the control and PLS groups with five different models. For Model 1, no adjustment for any factor was conducted. Model 2 was age-adjusted, and Model 3 was both age and educational level-adjusted. For Model 4, we adjusted for age, educational level, and total daily food intake. Finally, Model 5 included all the four potential confounders, including age, educational level, total daily food intake and total daily energy intake. We observed the increased ORs in Model 1 for the reference IQR1 for each factor, including fat, carotene, vitamin E-α, Vitamin E-(β+γ), magnesium, potassium, sodium and copper (Table 2), but no significant difference was found for other factors (Appendix A). After adjusting for age, educational level, total food intake per day and total energy intake per day, we still observed increased ORs from IQR2 to IQR4 in fat, vitamin E-(β+γ), magnesium, sodium and copper (all OR > 1 with *p* < 0.05) (Table 2). To reveal the impact of the nutrients on the NIANS group, we also adjusted for age, educational level, total daily food intake and total daily energy intake and found a similar tendency for fat, magnesium, and sodium with the results in PLS, as compared with the control group. We also found the NIANS-specific elevated factors of thiamin, riboflavin, and potassium (Appendix A). We also performed the analysis based on the tests for trend, modeling the factors into categories and conducting the one degree-of-freedom linear term analysis. We observed an association of PLS with increasing levels of fat (*P _for trend_* = 0.011), vitamin E-(β+γ) (*P _for trend_* = 0.003), magnesium (*P _for trend_* = 0.004), sodium (*P _for trend_* = 0.001) and copper (*P _for trend_* < 0.001) (Figure 2).

## 4. Discussion

To our knowledge, this is the first epidemiological case-control study to test the association between micronutrient intake and PLS susceptibility. Importantly, we found that the higher intake of fat, vitamin E-(β+γ), magnesium, sodium and copper was associated with a risk of PLS, and early intervention might prevent PLS.

A high dietary proportion of sugar, refined starches and saturated or trans-fatty acids may activate the innate immune system, which is associated with the excessive secretion of pro-inflammation cytokines [23,24,25]. Obesity impacts T cell activation, and more prominent Th17 inflammation occurs in obese humans with allergic disease [26]. Growing evidence suggests that rodents fed a high-fat diet exhibit an altered lipid composition in splenic T cells [27,28,29], leading to enhanced T cell proliferation and IL-2 pathway activation [29]. High fat intake might affect the PLS by altering the immune response and immunocyte activation.

In our study, the higher intake of vitamin E, especially the β and γ subclasses, was identified as the risk factor for PLS. Although the α-, β- and γ- tocopherols were reported to reduce inflammation via scavenging free radicals and nitrogen oxygen species [30,31], the excess dietary vitamin E has been shown to enhance cell-mediated and humoral immune responses, with increased lymphocyte proliferation, immunoglobulin levels, antibody responses, natural killer (NK) cell activity, and interleukin (IL)-2 production [32]. It has been reported that supplementation of γ tocotrienol can increase CD4+, CD8+ T-cells and natural killer cells but suppress Treg cells in peripheral blood [33]. Magnesium is a critical mineral which acts as a cofactor or activator in more than 300 enzymatic reactions [34]. Magnesium yields a protective effect by attenuating inflammation, improving insulin and glucose metabolism, normalizing the lipid profile and enhancing endothelium-dependent vasodilation [35]. Inconsistent with the literature, we found that higher magnesium intake levels increased the risk of PLS. It should be borne in mind that the increased intake value of magnesium might reflect the consumption of foods rich in magnesium but not the isolated effect, which means that the anti-inflammatory function of magnesium might be negated by the influence of foods rich in magnesium.

The high salt content in diets is suspected to be an independent factor that drives the significant burden and mortality rates [36]. The present study also found that higher salt intake was linked with increased PLS risk. An increasing body of evidence suggests that high salt consumption can stimulate the immune system, which results in a vigorous autoimmune response [37,38,39]. Excessive salt augments the differentiation of naïve T cells into Th17 [38]; a high salt diet can significantly increase the serum levels of INF-γ, TNF-α, IL-9, and IL-17A levels in young and healthy individuals [40]. Moreover, excessive salt was shown to exert a direct effect on the suppressive functions of Tregs [41]; the high salt levels that caused the TH17/Treg imbalance might be the reason that we found the phenomenon that a high sodium intake in the PLS group than the control group in the current study.

Copper plays a pivotal role in the immune system function; its deficiency might reduce T cell proliferation, which requires IL-2, and results in immune imbalance. In contrast, copper accumulates in inflamed tissues and contributes critically to host defense against bacterial infection [42]. The mean blood copper level in rheumatoid arthritis patients was significantly higher than control healthy patients; the blood copper level was positively correlated with the C-reactive protein level and erythrocyte sedimentation rate [43]. It is widely acknowledged that Wilson’s disease (WD) is an inherited disorder of chronic copper toxicosis; a study found higher plasma Th 1 cells, Th 2 cells, and Th17 cells in WD compared with control groups [44]. Meanwhile, IL-17 can induce the increased uptake of copper through STEAP4 [45]. Importantly, the cross-talk between the immune system and copper-associated signaling might accelerate the formation of chronic prostatitis.

To the best of our knowledge, this is the first study to investigate the different intake of nutrients among participants with or without PLS and the prodromal stage, NIANS. We recorded the daily diet information with the customized questionnaire via an offline one-to-one survey to increase the accuracy, then converted the information to the micronutrient amount based on the Chinese Food Composition Tables. Importantly, we converted continuous variables into categorical variables based on the IQR values, and calculated the *p*-Value for trend to identify the linear trend of altered ORs. Several limitations and shortcomings were present in this study. First, the effect of unknown confounders on our study findings cannot be completely ruled out. Therefore, we performed multiple logistic regression analyses to adjust for the impact of age, educational level, total daily food intake, and total daily energy intake in order to increase the robustness of our findings. Moreover, the study is based on the comparison of nutrient intake of PLS patients and normal subjects in a single locality center, and should be further verified in multiple regions and larger populations.

## 5. Conclusions

This is the first study to evaluate the distribution of nutrient intake among PLS patients, revealing that the higher intake of fat, vitamin E-(β+γ), magnesium, sodium and copper is associated with a risk of PLS. Further validation in large multi-center international cohorts is warranted to validate our findings.

## Figures and Tables

**Figure 1 nutrients-14-03675-f001:**
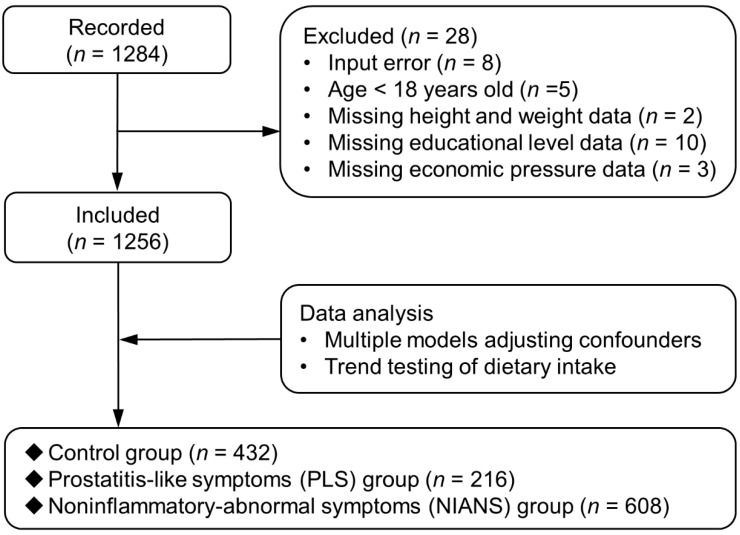
Flow chart of study.

**Figure 2 nutrients-14-03675-f002:**
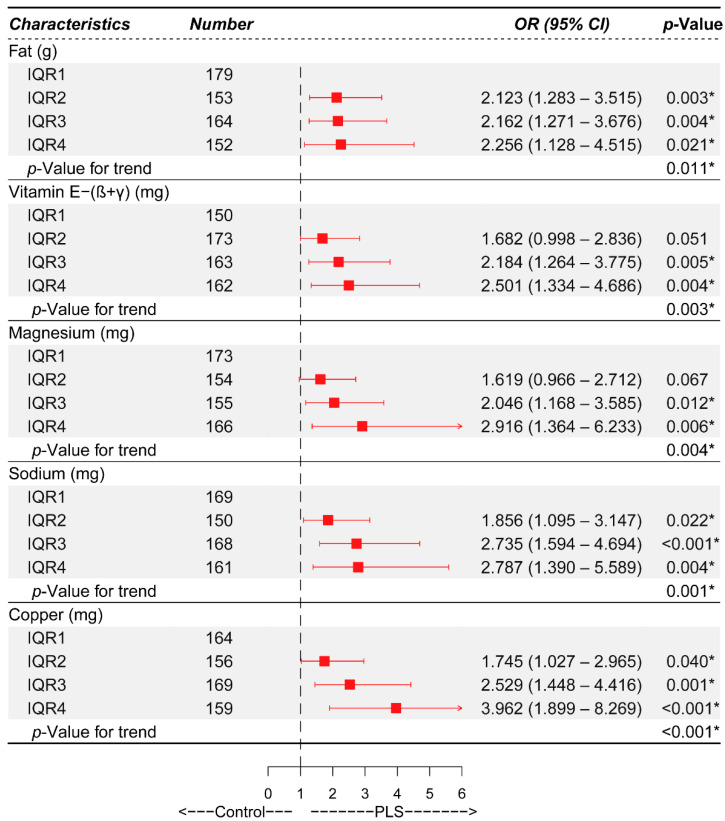
Elevated risk of prostatitis-like symptoms (or PLS) with higher intake of fat, vitamin E-(β+γ), magnesium, sodium and copper. *, *p*-Value < 0.05.

**Table 1 nutrients-14-03675-t001:** Characteristics of demographic and dietary intake in three groups.

Variables	Total (*n* = 1256)	Control Group (*n* = 432)	PLS Group (*n* = 216)	NIANS Group (*n* = 608)	*p*-Value ^1^	*p*-Value ^2^	*p*-Value ^3^
Age (y) *	21.0 (20.0, 26.0)	21.0 (20.0, 25.0)	26.0 (23.0, 38.0)	21.0 (20.0, 24.0)	**<0.001**	**<0.001**	0.612
BMI (kg/m^2^) *	22.6 (20.5, 25.0)	22.5 (20.5, 24.6)	22.9 (20.9, 25.5)	22.6 (20.4, 25.0)	0.316	0.129	0.646
Educational level ^#^					**<0.001**	**<0.001**	0.277
Senior high school or bachelor’s degrees	1093 (87.0%)	387 (89.6%)	165 (76.4%)	541 (89.0%)			
Junior high school or below	99 (7.9%)	27 (6.2%)	42 (19.4%)	30 (5.0%)			
Master’s degrees or above	64 (5.1%)	18 (4.2%)	9 (4.2%)	37 (6.0%)			
Economic pressure ^#^					0.091	0.404	**0.023**
None	404 (32.2%)	161 (37.3%)	66 (30.6%)	177 (29.1%)			
Mild	484 (38.5%)	148 (34.3%)	80 (37.0%)	256 (42.1%)			
Moderate	274 (21.8%)	90 (20.8%)	51 (23.6%)	133 (21.9%)			
Severe	94 (7.5%)	33 (7.6%)	19 (8.8%)	42 (6.9%)			
Total food intake per day (g) *	1107.0 (760.0, 1542.5)	1045.7 (720.0, 1502.5)	1150.6 (814.6, 1613.1)	1121.6 (789.7, 1545.1)	0.091	**0.044**	0.098
Total energy intake per day (KJ) *	5157.4 (3545.2, 7656.7)	4779.0 (3290.0, 7607.9)	5675.9 (3808.2, 7716.0)	5328.4 (3610.9, 7619.8)	**0.039**	**0.019**	0.055
Protein (g) *	64.7 (42.1, 99.7)	59.9 (38.5, 96.1)	68.8(48.7, 99.7)	65.4 (43.7, 105.0)	**0.024**	**0.017**	**0.023**
Fat (g) *	33.8 (21.4, 56.0)	32.4 (18.0, 54.5)	37.6 (24.4, 54.4)	35.1 (22.8, 57.0)	**0.022**	**0.031**	**0.013**
Carbohydrate (g) *	162.9(108.5, 240.9)	151.6 (103.4, 228.8)	173.6 (125.0, 244.8)	163.0 (108.7, 240.0)	0.065	**0.019**	0.312
Dietary fiber (g) *	7.24 (4.4, 12.2)	6.7 (3.8, 11.7)	8.5 (5.3, 13.2)	7.2 (4.3, 12.4)	**0.002**	**<0.001**	0.124
Cholesterol (mg) *	346.2 (156.9, 519.8)	311.0 (137.7, 482.1)	385.5(180.6, 545.3)	356.3 (162.9, 539.3)	**0.016**	**0.006**	**0.044**
Ash (g) *	8.0 (5.2, 12.1)	7.3 (4.8, 11.5)	8.7 (5.9, 12.6)	8.1 (5.3, 12.0)	**0.007**	**0.003**	**0.025**
Vitamin A (µg) *	494.9(287.6, 802.9)	481.5 (261.0, 770.5)	495.5 (323.7, 847.4)	501.0 (296.8, 804.7)	0.120	0.080	0.082
Carotene (µg) *	1465.3 (539.3, 2868.0)	1312.8 (459.3, 2897.9)	1668.1 (873.0, 3068.7)	1440.1 (541.7, 2775.6)	**0.046**	**0.022**	0.691
Retinol (µg) *	173.2 (98.1, 272.4)	158.9 (89.8, 258.8)	159.3 (88.9, 248.9)	189.4 (108.0, 292.0)	**0.003**	0.841	**0.002**
Thiamin (mg) *	0.6 (0.44, 0.9)	0.5 (0.3, 0.9)	0.6 (0.4, 0.9)	0.6 (0.4, 0.9)	0.088	0.095	**0.042**
Riboflavin (mg) *	0.8 (0.5, 1.2)	0.8 (0.5, 1.2)	0.85 (0.6, 1.2)	0.8 (0.6, 1.3)	**0.022**	0.060	**0.008**
Vitamin C (mg) *	53.9 (28.2, 94.4)	51.9 (24.5, 91.3)	62.8 (33.5, 107.9)	52.5 (28.2, 92.2)	**0.014**	**0.005**	0.527
Vitamin E [total (mg)] *	10.4 (5.9, 18.7)	9.9(5.3, 17.3)	13.0 (7.0, 20.6)	10.4 (6.0, 18.5)	**0.005**	**0.001**	0.154
Vitamin E-α (mg) *	4.6 (2.7, 7.7)	4.2 (2.4, 7.2)	4.9 (3.2, 8.0)	4.7 (2.7, 7.7)	**0.008**	**0.002**	0.091
Vitamin E-(β+γ) (mg) *	3.4 (1.8, 7.0)	3.1 (1.6, 6.4)	4.1 (2.4, 7.9)	3.4 (1.7, 7.0)	**0.006**	**0.001**	0.286
Vitamin E-δ (mg) *	1.8 (0.9, 3.5)	1.8 (0.9, 3.2)	1.9 (1.1, 3.9)	1.8 (0.9, 3.5)	0.174	0.062	0.631
Calcium (mg) *	522.6 (323.2, 782.3)	492.6 (296.7, 760.3)	519.3 (332.7, 747.9)	557.5 (341.0, 798.6)	0.119	0.335	**0.042**
Zinc (mg) *	9.3 (6.2, 14.5)	8.6(5.7, 13.7)	10.1 (6.9, 14.3)	9.6 (6.4, 15.1)	0.060	**0.038**	**0.049**
Magnesium (mg) *	219.1 (143.6, 331.7)	209.0 (131.3, 319.8)	242.8 (162.3, 368.2)	221.0 (146.9, 326.4)	**0.005**	**0.001**	0.072
Selenium (mg) *	34.4 (21.4, 55.5)	31.3(19.5, 51.9)	37.6 (25.4, 58.8)	35.3 (21.5, 55.5)	**0.017**	**0.006**	**0.047**
Phosphorus (mg) *	948.4 (626.2, 1377.8)	868.98 (604.2, 1326.3)	996.73 (669.9, 1308.7)	973.64 (647.4, 1409.3)	0.052	0.078	**0.022**
Potassium (mg) *	1556.9 (1013.5, 2456.7)	1438.6 (894.0, 2382.2)	1688.3 (1108.7, 2475.3)	1589.4 (1046.1, 2467.0)	**0.010**	**0.007**	**0.016**
Sodium (mg) *	565.6 (342.4, 958.0)	521.1 (298.9, 882.7)	659.3 (437.4, 1013.1)	558.4 (356.3, 959.4)	**<0.001**	**<0.001**	**0.038**
Iron (mg) *	16.1 (10.2, 24.7)	14.9 (9.6, 23.8)	17.3 (11.6, 24.8)	16.3(10.5, 25.1)	**0.036**	**0.019**	**0.045**
Copper (mg) *	1.3 (0.8, 2.1)	1.2 (0.7, 1.9)	1.5 (1.0, 2.3)	1.3 (0.8, 2.2)	**<0.001**	**<0.001**	0.070
Manganese (mg) *	3.7 (2.3, 5.6)	3.4 (2.2, 5.4)	4.1 (2.7, 5.7)	3.8 (2.3, 5.6)	**0.017**	**0.005**	0.107

***Note****:* Bold font indicates *p*-Value < 0.05. * Quantitative data, non-normal distribution, presented in the form of median (IQR), and Kruskal-Wallis (K-W) test for the comparison of groups’ differences; **^#^** Qualitative data, presented as quantity (percentage), and Chi-square test for the comparison of groups’ differences. ^1^ K-W/Chi-square test between the three groups; ^2^ K-W/Chi-square test between control and PLS group; ^3^ K-W/Chi-square test between control and NIANS group. ***Abbreviation***: PLS, prostatitis-like symptoms; NIANS, noninflammatory-abnormal symptoms; Ref., reference; *n*, number; y, years, BMI, body mass index; g, gram; kg, kilogram; m, meter; KJ, kilojoule; mg, milligram; ug, microgram; IQR, interquartile range.

**Table 2 nutrients-14-03675-t002:** Multiple logistic regression models were constructed by adding confounding factors gradually to identify nutrient intake associated with PLS in the daily diet.

		Model 1		Model 2		Model 3		Model 4		Model 5	
Variables	No.	OR (95% CI)	*p*-Value	OR (95% CI)	*p*-Value	OR (95% CI)	*p*-Value	OR (95% CI)	*p*-Value	OR (95% CI)	*p*-Value
Fat (g) *											
IQR1	179	Ref.		Ref.		Ref.		Ref.		Ref.	
IQR2	153	2.181 (1.351, 3.522)	**0.001**	2.053 (1.256, 3.355)	**0.004**	2.037 (1.244, 3.336)	**0.005**	2.102 (1.271, 3.476)	**0.004**	2.123 (1.283, 3.515)	**0.003**
IQR3	164	2.282 (1.425, 3.653)	**0.001**	1.928 (1.187, 3.133)	**0.008**	1.933 (1.186, 3.150)	**0.008**	2.054 (1.218, 3.466)	**0.007**	2.162 (1.271, 3.676)	**0.004**
IQR4	152	1.807 (1.112, 2.937)	**0.017**	1.681 (1.023, 2.762)	**0.041**	1.690 (1.026, 2.786)	**0.039**	1.927 (1.015, 3.655)	**0.045**	2.256 (1.128, 4.515)	**0.021**
Carotene (µg) *											
IQR1	163	Ref.		Ref.		Ref.		Ref.		Ref.	
IQR2	157	1.812 (1.116, 2.944)	**0.016**	1.630 (0.993, 2.674)	0.053	1.587 (0.964, 2.612)	0.069	1.582 (0.959, 2.607)	0.072	1.571 (0.953, 2.591)	0.077
IQR3	159	1.979 (1.223, 3.202)	**0.005**	1.603 (0.974, 2.638)	0.064	1.621 (0.983, 2.673)	0.059	1.606 (0.963, 2.679)	0.069	1.580 (0.945, 2.641)	0.081
IQR4	169	1.705 (1.056, 2.753)	**0.029**	1.371 (0.835, 2.252)	0.212	1.408 (0.855, 2.318)	0.178	1.375 (0.775, 2.439)	0.276	1.331 (0.744, 2.383)	0.335
Vitamin E-α (mg) *											
IQR1	165	Ref.		Ref.		Ref.		Ref.		Ref.	
IQR2	165	1.960 (1.212, 3.171)	**0.006**	1.738 (1.061, 2.844)	**0.028**	1.780 (1.084, 2.921)	**0.023**	1.789 (1.083, 2.955)	**0.023**	1.803 (1.091, 2.980)	**0.022**
IQR3	157	2.012 (1.238, 3.269)	**0.005**	1.766 (1.074, 2.904)	**0.025**	1.805 (1.093, 2.980)	**0.021**	1.825 (1.074, 3.101)	**0.026**	1.844 (1.084, 3.135)	**0.024**
IQR4	161	1.882 (1.159, 3.056)	**0.011**	1.436 (0.866, 2.381)	0.161	1.486 (0.893, 2.471)	0.127	1.519 (0.825, 2.794)	0.179	1.617 (0.862, 3.034)	0.134
Vitamin E-(β+γ) (mg) *											
IQR1	150	Ref.		Ref.		Ref.		Ref.		Ref.	
IQR2	173	1.719 (1.037, 2.848)	**0.036**	1.591 (0.952, 2.657)	0.076	1.583 (0.946, 2.650)	0.080	1.640 (0.974, 2.761)	0.063	1.682 (0.998, 2.836)	0.051
IQR3	163	2.384 (1.444, 3.936)	**0.001**	1.958 (1.170, 3.276)	**0.011**	1.946 (1.159, 3.268)	**0.012**	2.095 (1.217, 3.609)	**0.008**	2.184 (1.264, 3.775)	**0.005**
IQR4	162	2.471 (1.497, 4.079)	**<0.001**	1.892 (1.124, 3.182)	**0.016**	1.927 (1.143, 3.251)	**0.014**	2.215 (1.210, 4.055)	**0.010**	2.501 (1.334, 4.686)	**0.004**
Magnesium (mg) *											
IQR1	173	Ref.		Ref.		Ref.		Ref.		Ref.	
IQR2	154	1.695 (1.042, 2.757)	**0.033**	1.464 (0.890, 2.408)	0.134	1.457 (0.885, 2.401)	0.139	1.568 (0.938, 2.619)	0.086	1.619 (0.966, 2.712)	0.067
IQR3	155	1.934 (1.195, 3.129)	**0.007**	1.635 (0.999, 2.676)	0.051	1.619 (0.986, 2.657)	0.057	1.858 (1.079, 3.200)	**0.025**	2.046 (1.168, 3.585)	**0.012**
IQR4	166	2.250 (1.406, 3.601)	**0.001**	1.699 (1.041, 2.774)	**0.034**	1.693 (1.034, 2.770)	**0.036**	2.295 (1.148, 4.587)	**0.019**	2.916 (1.364, 6.233)	**0.006**
Potassium (mg) *											
IQR1	174	Ref.		Ref.		Ref.		Ref.		Ref.	
IQR2	157	1.909 (1.181, 3.086)	**0.008**	1.616 (0.987, 2.647)	0.057	1.617 (0.985, 2.656)	0.057	1.652 (0.994, 2.746)	0.053	1.670 (1.004, 2.779)	**0.048**
IQR3	159	2.257 (1.404, 3.627)	**0.001**	1.925 (1.185, 3.130)	**0.008**	1.929 (1.183, 3.147)	**0.008**	2.018 (1.174, 3.471)	**0.011**	2.064 (1.196, 3.560)	**0.009**
IQR4	158	1.789 (1.105, 2.895)	**0.018**	1.400 (0.850, 2.306)	0.186	1.442 (0.873, 2.381)	0.153	1.595 (0.773, 3.290)	0.206	1.714 (0.815, 3.603)	0.155
Sodium (mg) *											
IQR1	169	Ref.		Ref.		Ref.		Ref.		Ref.	
IQR2	150	2.045 (1.234, 3.391)	**0.006**	1.751 (1.043, 2.940)	**0.034**	1.743 (1.037, 2.929)	**0.036**	1.823 (1.077, 3.086)	**0.025**	1.856 (1.095, 3.147)	**0.022**
IQR3	168	2.906 (1.790, 4.720)	**<0.001**	2.418 (1.470, 3.975)	**0.001**	2.346 (1.421, 3.871)	**0.001**	2.568 (1.509, 4.370)	**0.001**	2.735 (1.594, 4.694)	**<0.001**
IQR4	161	2.359 (1.440, 3.864)	**0.001**	1.878 (1.129, 3.124)	**0.015**	1.868 (1.120, 3.113)	**0.017**	2.279 (1.200, 4.328)	**0.012**	2.787 (1.390, 5.589)	**0.004**
Copper (mg) *											
IQR1	164	Ref.		Ref.		Ref.		Ref.		Ref.	
IQR2	156	1.688 (1.020, 2.793)	**0.042**	1.510 (0.904, 2.524)	0.116	1.527 (0.913, 2.555)	0.107	1.701 (1.002, 2.886)	**0.049**	1.745 (1.027, 2.965)	**0.040**
IQR3	169	2.191 (1.347, 3.563)	**0.002**	1.912 (1.164, 3.141)	**0.010**	1.867 (1.134, 3.075)	**0.014**	2.261 (1.315, 3.888)	**0.003**	2.529 (1.448, 4.416)	**0.001**
IQR4	159	2.826 (1.735, 4.602)	**<0.001**	2.073 (1.245, 3.453)	**0.005**	2.091 (1.253, 3.488)	**0.005**	3.212 (1.604, 6.431)	**0.001**	3.962 (1.899, 8.269)	**<0.001**

***Note***: Bold font indicates *p*-Value < 0.05. * Quantitative data, non-normal distribution, presented in the form of median (IQR), and Kruskal-Wallis (K-W) test for the comparison of groups’ differences; Model 1, Crude model. Model 2, Adjusted for age. Model 3, Adjusted for age, educational level. Model 4, Adjusted for age, educational level, total food intake per day (g). Model 5, Adjusted for age, educational level, total food intake per day (g), total energy intake per day (KJ). *Subclassification (based on IQR):* subtype 1, 0 ≤ element quantification ≤ 0.25; subtype 2, 0.25 < element quantification ≤ 0.50; subtype 3, 0.50 < element quantification ≤ 0.75; subtype 4, 0.75 < element quantification ≤ 1. ***Abbreviation***: PLS, prostatitis-like symptoms; No, number; y, years, BMI, body mass index; kg, kilogram; m, meter; KJ, kilojoule; mg, milligram; ug, microgram; IQR, interquartile range.

## Data Availability

All the data and materials mentioned in the manuscript are available.

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
