# Peer review of "Higher Intake of Fat, Vitamin E-(β+γ), Magnesium, Sodium, and Copper Increases the Susceptibility to Prostatitis-like Symptoms: Evidence from a Chinese Adult Cohort"

_nutrients, 2022, doi:10.3390/nu14183675_

Round 1

Reviewer 1 Report

The authors studied the effect of dietary micronutrients on prevalence of Prostate-like Symptoms (PLS) in a population cohort prospectively recruited from local hospitals, universities, and communities, consecutively, in an area of China province. They concluded that the intake of fat, vitamin E-(β+γ), magnesium, sodium and cuprum was associated with a higher risk of PLS.

The study is relevant, but the manuscript shows lots of defects and inaccuracies mostly related to the English language and the scientific writing. There are statements that are not understandable to English speaking readers. Moreover, there are defects and mistakes in all sections mostly related to scientific writing methodology. As a whole, the paper cannot be considered publishable the way it is.

The authors should submit it to a professional scientific writer and make a deep review.

Introduction

The definition of PLS is missing.

36-38: statement is not clear. I cannot understand the relationship between micronutrients and PLS;

41-48: introduction is not the place for detailed explanations. Any detailed explanation should be discussed in the discussion section.

56: how this statement is supported? In scientific writing this kind of statement must be supported.

60-62: This is a conclusion statement. The introduction it is not its place.

Material and methods

The M&M section is not assessable because of lacking supplementary material declared in the text.

This session should be written in a way that the readers could reproduce the same study in their institutions.

The variables description is insufficient. I cannot understand the variables in your field. You should clearly specify the dependent and independent variables that you included in the stats as well as their justification.

Results

I can see 3.1 and 3.3 subchapters. Subchapter 3.2 is missing.

112-116: this is a M&M statement. Results section it is not his place. However, how did you select the three study groups is not clear. I understood that the PLS group has pain domain at least 4 or above (with any NIH-CPSI?); the control group should have NIH-CPSI score zero; the NIANS should have any NIH-CPSI score but pain domain zero. It should be explained in English.

116: You should specify the number and not make the reader to do the calculation.

145-153: If you have already shown these stats data in the tables, you do not need to report them in the text. It makes the reading very hard.

Discussion

Mostly not understandable to normal international readers. The authors discussed also topics not included in the present study.

Reviewer 2 Report

This study attempts to corelate the dietary intake of macro- and micronutrients with prostate noninflammatory abnormal symptoms (NIAANS) and more advanced prostatitis-like symptoms (PLS) in a cohort of adult Chinese males (n=1256). The control group (no symptoms) included 432 subjects, NIANS group – 608 subjects, and PLS group – 216 subjects. The subjects were interviewed using customized (not “home-made” or “self-made”) dietary questionnaire and demographic data were also collected. Statistical analysis suggested that higher intakes of fat, vitamin E- (β + γ), as well as Mg, Na and Cu, were associated with higher risk of PLS.

The article does not compare the median values and inter-quartile ranges with any recommended intakes for nutrients (RDA, DRI, adequate intake) to show that the intakes of some subjects were really excessive (as in the Title). Both educational level and economic pressure categories are not explained. What is “Moisture” intake in Table 1 (drinking water? fluids? water content in foods? all together?)? There is no need to repeat all the numerical information from Table 2 in the text (lines 145-154). The Discussion concentrates too much on the problems associated with inadequate intake of investigated nutrients instead of on the consequences of high (excessive?) intake.

The article is written in very deficient English, with many errors of style, grammar and spelling. It is impossible to enumerate them all, but here are a few examples.

Some mineral names are occasionally in Latin version (copper – cuprum), other are always in Latin (kalium, ferrum), or English (sodium, manganese). All of them should be always in English.

Inappropriate words or wrong grammatical form are abundant:  “China food composition table” instead of “Chinese…“ (line 19, 88), “psychosocial status “ (line 41), “hot pot in mechanism study” (line 42),”brand new understanding” (line 61-62) ”collected participants” instead of “enrolled” (line67), “conditioning on” instead of “adjusted for” (line 104), “bias” instead of “confounders” (line 139), “Identify” instead of “Identification” (line 133), “inflammatory” instead of “inflammation” (line 198, 206), “associated” instead of “association” (line 213), and many other.

Repetitious careless writing, as in line 59-60: “… the role of micronutrients, such as microelement, micronutrients, vitamins,”. Instead of “revealed the brand new understanding” it should be “found”.

Typing mistakes, as in Figure 1, The sign after “Age” should be > (more), not < (less). “Vitamin” is misspelled twice in Table 1.

It is difficult to understand the meaning of some sentences, as on line 213-215 (“pointed out in about 50 years of age”? Was it pointed out 50 years ago, but quoted by [43] more recently?) The sentence proceeds in such convoluted and meaningless way (“and recent studies have provided new insights of salt to health, the pro-inflammatory ways”).

Figure 2 is more of a table than figure. As the perpendicular line denotes reference (control) (OR 1 on the bottom scale), it is not necessary to write Ref. in the third column, next to the perpendicular line. The title of this figure should be “Elevated risk of prostatitis-like symptoms (or PLS) with higher intake of fat, vitamin E- (β + γ), magnesium, sodium and copper.” The capitalization must be consistent.

In Table 1 some numbers should be rounded, so they will take less space and make the table more readable (example: 1157.36 to 1157.4 or 1157).

In summary, the article cannot be published in present shape, it must be carefully reviewed and corrected by the authors, as well as edited by a professional English editor.

Reviewer 3 Report

1. The aim of the study must be mentioned as a separate paragraph at the end of the introduction.

2. The English language needs to be improved. It should be carefully double checked and improved. The scientific reports must be written professionally. Please double check the manuscript.

3. The whole manuscript must be prepared regarding the given instruction of the journal. The references should be checked one more time.

4. Please explain and clarify this subject has been selected for study and specify the main aim of the current study in 2 lines at the end of introduction.

5.The discussion should improve. Very good results are achieved and they could be presented in better ways as well.

6.The unites must be regarding the standard symbols in whole the manuscript, whole the manuscript must be double checked.

7. Some of tables could be presented as figures instead. This might increase the quality of the presentation of the achieved results.

8. Very recent articles are recommended to be cited in the manuscript. Very good results have been achieved. The discussion should be expanded.

Round 2

Reviewer 1 Report

I reviewed the paper. All our suggestions were well incorporated and the paper now is well written in reasonable scientific English

Author Response

Thanks for your approve of our study in current version!

Reviewer 2 Report

The amended version of this article is better than the original, but it still requires many corrections. The authors disregarded some suggestions of the reviewer and did not address many issues. The very title should be corrected by replacing “Excessive intake…” by “Higher intake…”. The reported intakes (Table 1) are not excessive according to any dietary recommendations. Similarly, “higher intake” should be used on lines 26, 193, 203, 258 instead of “intake”. The intake of these micronutrients is necessary for health!

Below are more corrections and questions:

Abstract. Line 17. It should be “collected information on…”, or “enrolled 1,284 participants”. Line 21. It should be “vitamin E-(β+γ)”.

Introduction. Line 33. It should be “difficult urination”. Line 36. It should be “low-vegetable or meat-rich diets”. Line 54. It should be “reactive”.

Materials and Methods. Line 72. Remove “study”. Line 75 – comma after 4. Line 82. It should be “was quantified”. Line 91. It should be “customized”, not “self-made”, or “homemade” (line 109, 243), as pointed before. It should be “seafood” not “aquatic products”, unless you specify fish, seaweed, etc.

The level of education, divided into the categories on lines 89-90 (also in Table 1), is cryptic. It should be explained, as pointed before, that Chinese education system involves 6 years of primary school, 3 years of lower secondary school (junior high school), 3 years of upper secondary school (high school). What is “undergraduate” and “graduate”? Standard university in China takes 4 years (?).

 Line 94. Average daily intake of foods cannot be in milligrams, but grams (also line 106).

Line 96-98. Remove “in the form of how many times” and “as how many times”. It should be ”low-intake food frequency was recorded per month” and “(average intake less than once per month)”.

Line 102-103. It should be “if the subject recalled 1-2 times per week, it was recorded as 1.5 times per week.” Line 104. It should be “…bias and accurately quantify each dietary intake [22].”

Line 105-106. It should be “the content of each per 100 g of edible part in commonly consumed food was compiled, based…”

Line 119 -120. It should be “between nutrients and PLS.” Fat is not a micronutrient.  “Data for each nutrient were divided into four quartiles; IOR1 was chosen…”. Remove “to reveal the impact of the other three groups.”

Line 125. See the questions about education system above.

Line 128-129. Remove “to uncover the real relational degree between micronutrients and PLS.”

Results. Table 1. Replace “between” in the title with “in three study groups”. What is the meaning of the asterisk next to each line? Can it be removed? Correct the description of each educational level (see above). Remove the data for moisture in food, they are not important in this context. Use symbol µ for µg. What is the meaning of Subclassification and subtypes 1,2, 3, 4 in the legend under the table? It does not appear anywhere in the table or text. The same is true for Table 2 (asterisks, subclassification, symbol). Table 2 title – replace “high-risk elements” with “nutrient”.

Line 172 -175. Replace “micronutrients” with “nutrients”. Line 173. It should be “and found a similar tendency for fat” The rest of the sentence is confusing and ungrammatical: “compared with PSL vs. control, and NIANS-specific factors” (?).

Line 177. Remove “striking”. It should be “association of PSL with….”

Discussion.  Line 210-211. Remove “in human organism”.

Line 216-218. It seems that the authors reversed cause and effect in this sentence (after semicolon). How an imbalance in TH17/Treg may cause higher sodium intake? Sodium intake in this study was low in all groups by WHO standards (recommended < 2 g/day), American Heart association (recommended <2.3 d/day, ideal < 1.5 b/day), and compared to average intake in China (5.6 g/day). Sodium in Chinese diet comes also from sodium glutamate, in addition to sodium chloride (salt).

Line 230. Put a comma after IL-2. Line 232. Remove ”In this respect, a study found that…”

Line 241-242. It should be “… different intake of nutrients…” (or ‘micro- and macronutrients”).

Line 252-253. It should be “…comparison of nutrients intake in PLS patients and normal subjects in a single locality…”

Line 254-255. This sentence is totally superfluous. What prevents the authors to make this comparison in the discussion, as was suggested before by the reviewer?

Line 257. It should be “distribution of nutrient intake among PLS patients”.

References. Most journal titles are abbreviated, but some are not (6, 17, 19, 25, 37). It should be uniform.

In summary, the paper still needs editing and it should address the points raised  in this and the previous review to be publishable.

Reviewer 3 Report

I think the manuscript was implemented after the authors' review. No further comment

Author Response

(The authors gave the same response as above.)
